# A Meta-Analytical Comparison of the Effects of Small-Sided Games vs. Running-Based High-Intensity Interval Training on Soccer Players’ Repeated-Sprint Ability

**DOI:** 10.3390/ijerph18052781

**Published:** 2021-03-09

**Authors:** Filipe Manuel Clemente, Rodrigo Ramirez-Campillo, José Afonso, Hugo Sarmento, Thomas Rosemann, Beat Knechtle

**Affiliations:** 1Escola Superior Desporto e Lazer, Instituto Politécnico de Viana do Castelo Rua Escola Industrial e Comercial de Nun’Álvares, 4900-347 Viana do Castelo, Portugal; 2Instituto de Telecomunicações, Delegação da Covilhã, 1049-001 Lisboa, Portugal; 3Quality of Life and Wellness Research Group, Human Performance Laboratory, Department of Physical Activity Sciences, Universidad de Los Lagos, Osorno 5290000, Chile; r.ramirez@ulagos.cl; 4Centro de Investigación en Fisiología del Ejercicio, Facultad de Ciencias, Universidad Mayor, Santiago 7500000, Chile; 5Centre for Research, Education, Innovation and Intervention in Sport, Faculty of Sport of the University of Porto, 4200-450 Porto, Portugal; jafonsovolei@hotmail.com; 6Research Unit for Sport and Physical Activity, Faculty of Sport Sciences and Physical Education, University of Coimbra, Coimbra, 3004-531 Coimbra, Portugal; hg.sarmento@gmail.com; 7Institute of Primary Care, University of Zurich, 8091 Zurich, Switzerland; thomas.rosemann@usz.ch; 8Medbase St. Gallen Am Vadianplatz, 9001 St. Gallen, Switzerland

**Keywords:** football, athletic performance, drill-based games, interval training, repeated sprint

## Abstract

This systematic review with a meta-analysis was conducted to compare the effects of small-sided games (SSGs)-based interventions with the effects of running-based high-intensity interval training (HIIT) interventions on soccer players’ repeated sprint ability (RSA). The data sources utilized were Web of Science, Scopus, SPORTDiscus, and PubMed. The study eligibility criteria were: (i) parallel studies (SSG-based programs vs. running-based HIIT) conducted in soccer players with no restrictions on age, sex, or competitive level; (ii) isolated intervention programs (i.e., only SSG vs. only running-based HIIT as individual forms) with no restrictions on duration; (iii) a pre–post outcome for RSA; (iv) original, full-text, peer-reviewed articles written in English. An electronic search yielded 513 articles, four of which were included in the present study. There was no significant difference between the effects of SSG-based and HIIT-based training interventions on RSA (effect size (ES) = 0.30; *p* = 0.181). The within-group analysis revealed no significant effect of SSG-based training interventions (ES = −0.23; *p* = 0.697) or HIIT-based training interventions (ES = 0.08; *p* = 0.899) on RSA. The meta-comparison revealed that neither SSGs nor HIIT-based interventions were effective in improving RSA in soccer players, and no differences were found between the two types of training. This suggests that complementary training may be performed to improve the effects of SSGs and HIIT. It also suggests that different forms of HIIT can be used because of the range of opportunities that such training affords.

## 1. Introduction

Small-sided games (SSGs) are adjusted formats of play that are often used in soccer training to develop a specific tactical/technical attribute [1] while intensifying some load parameters [2,3]. Typically, SSGs are designed according to different task constraints, which act concurrently to promote changes in the tactical/technical, physiological/physical, and psychological dimensions of players [3,4,5]. Naturally, the changes promoted by these games are influenced by how the constraints interact with each other [1]. Such constraints include the format of play (i.e., the number of players involved and the numerical relationships), pitch configuration (i.e., pitch size and shape), scoring method (e.g., with or without goalkeeper, goals or no goals), action restrictions (e.g., limited number of ball touches, limited movements), and tactical/strategical instructions (e.g., type of defensive marking, type of attack) [6,7]. These constraints are related to the structure of the game. However, another important constraint is related to the training regimen (e.g., work duration, recovery duration, work-to-rest ratio) [3].

Usually, SSGs are prescribed with regimens similar to those recommended for long-interval high-intensity interval training (HIIT) sessions [8]. The internal load demands imposed by SSGs and running-based long-interval HIIT are also similar, and this has been considered one of the reasons for using SSG as a replacement for running-based HIIT; SSGs also have the advantage of developing tactical/technical issues [2,9]. In a meta-comparison between SSGs and conventional endurance training (in which running-based HIIT was included), the effects on aerobic performance were similar between both types of training (trivial differences), and within-group analyses of both revealed beneficial effects [10]. However, aerobic performance is just one of the many physical qualities that players must develop to support the demands of the game.

Soccer is characterized by its intermittent nature, in which low-to-moderate intensity activities are interspaced with highly demanding activities in which explosive actions and repeated high exertion occur based on the context of the game [11]. Among other factors, repeated sprint ability (RSA) is a determinant physical component since the capacity to sustain repeated high-intensity efforts is often needed for different periods of a match and is associated with overall match performance [12]. Due to the complexity of RSA, it has several limiting factors (e.g., muscular factors, neural factors) [12]. Naturally, the training process is one of the variables that may alter RSA, especially considering energy supply, hydrogen accumulation, and muscle activation [13]. The training approaches that are implemented to develop RSA include repeated sprint training, sprint training, SSGs, and resistance training [13].

In soccer, the optimization of the training time is crucial. Therefore, it is important to understand whether drill-based exercises (e.g., SSGs) can develop RSA to a similar extent as other forms of exercise (e.g., running-based HIIT) and whether they are significantly beneficial for soccer players. Such an understanding (SSG vs. running-based HIIT) will help define a practical application for the soccer field.

Additionally, a systematic review and meta-analysis will help summarize the main training protocols and parallel studies that have compared SSGs and running-based HIIT, with a focus on their effects on RSA. Although two meta-analyses of SSGs have been carried out recently [10,14], one did not consider RSA [10], and the other only included young players and did not objectively compare running-based HIIT with SSGs [14]. Thus, the need remains for a systematic review and meta-analysis that consolidates evidence about the effects of these forms of training on the RSA of soccer players. The purpose of this systematic review and meta-analysis was to compare the effects of SSG-based interventions vs. the effects of running-based HIIT interventions on soccer players’ RSA.

## 2. Materials and Methods

This study followed the Cochrane Collaboration guidelines [15] and the Preferred Reporting Items for Systematic Reviews and Meta-Analyses (PRISMA) guidelines [16]. The protocol was registered with the International Platform of Registered Systematic Review and Meta-Analysis Protocols with the number INPLASY202080129.

### 2.1. Information Sources

A comprehensive computerized search of the following electronic databases was performed: (i) Web of Science; (ii) Scopus; (iii) SPORTDiscus; (iv) PubMed. The searching process for relevant publications had no restriction regarding the year of publication and included articles retrieved until 1 September 2020. The following search strings were employed: (“soccer” OR “football”) AND (“small-sided games” OR “drill-based games” OR “sided-games” OR “SSG” OR “conditioned games” OR “small-sided and conditioned games” OR “reduced games” OR “play formats”) AND (“sprint”).

The following inclusion criteria were established: (i) parallel randomized studies (SSG-based programs vs. running-based HIIT) conducted in soccer players with no restriction of age, sex, or competitive level; (ii) isolated intervention programs (i.e., only SSG vs. only running-based HIIT as discrete forms) with no restrictions on duration; (iii) a pre–post outcome for RSA; (iv) original, peer-reviewed articles written in English that provided the full text.

Studies were excluded on the basis that they (i) were observational analytic designs; (ii) included other sports; (iii) used SSG or running-based HIIT combined with other training methods or between them (e.g., SSG + running based-HIIT); (iv) were conducted in recreational soccer (e.g., healthy population but not soccer players) or physical education contexts; (iv) were review articles, letters to the editor, errata, invited commentaries, or conference abstracts.

### 2.2. Data Extraction

An Excel spreadsheet was designed (Microsoft Corporation, Redmond, WA, USA) to process the data extraction [17]. Two of the authors performed the data extraction (F.M.C. and H.S.). Disagreements about study eligibility were solved in discussions between the authors. Full-text articles that were excluded were recorded with reasons for exclusion. All of the records were stored in the spreadsheet.

### 2.3. Data Items

The outcomes chosen for this systematic review and meta-analysis included RSA measured through field-based tests. The RSA was collected based on the mean time (s), mean power (W), or total time (s) in a series of multiple sprints. Additionally, the following information was extracted from the included studies: (i) the number of participants (n), age (years), competitive level (if available), and sex; (ii) the SSGs format and pitch size (if available); (iii) the period of intervention (number of weeks) and number of sessions per week (n/w); (iv) the regimen of intervention (work duration, work intensity, modality, relief duration, relief intensity, repetitions and series, and between-set recovery).

### 2.4. Assessment of Methodological Quality

To assess the methodological quality of the included articles, the methodological index for non-randomized studies (MINORS) was used [18]. Twelve items were analyzed, in which there were 0 represented cases of no report, 1 case reported but inadequate, and 2 cases reported and adequate. Two of the authors (F.M.C. and H.S.) independently scored the articles. Any disagreements were resolved through discussion. The inter-observer analysis was conducted using a Kappa correlation test. An agreement level of k = 0.91 was obtained.

### 2.5. Summary Measures

The analysis and interpretation of results in this systematic review and meta-analysis were conducted only in the case that at least three study groups provided baseline and follow-up data for RSA [19,20,21]. Means and standard deviations for RSA were converted to Hedges’ *g* effect size (ES). The inverse-variance random-effects model for meta-analyses was used because it allocates a proportionate weight to trials based on the size of their individual standard errors [22] and enables analysis while accounting for heterogeneity across studies [23]. The ESs were presented alongside 95% confidence intervals (CIs) and were interpreted using the following thresholds [24]: <0.2, trivial; 0.2–0.6, small; >0.6–1.2, moderate; >1.2–2.0, large; >2.0–4.0, very large; >4.0, extremely large. All analyses were carried out using the Comprehensive Meta-Analysis program (version 2; Biostat, Englewood, NJ, USA).

### 2.6. Synthesis of Results

To estimate the degree of heterogeneity between the included studies, the percentage of total variation across the studies due to heterogeneity was used to calculate the *I*^2^ statistic [25]. Low, moderate, and high levels of heterogeneity correspond to *I*^2^ values of <25%, 25–75%, and >75%, respectively [25,26].

### 2.7. Risk of Bias Across Studies

The extended Egger’s test [27] was used to assess the risk of bias across the studies. In the case of bias, Duval and Tweedie’s trim and fill method was conducted.

## 3. Results

### 3.1. Study Identification and Selection

The searching of databases identified a total of 513 titles. These studies were then exported to the reference manager software EndNote^TM^ X9 (Clarivate Analytics, Philadelphia, PA, USA). Duplicates (249) were subsequently removed, either automatically or manually. The remaining 264 articles were screened for their relevance based on titles and abstracts, resulting in the removal of a further 242 studies. The full texts of the remaining 22 articles were examined diligently. After reading the full texts, a further 18 studies were excluded due to a number of reasons (Figure 1). The four studies included in the meta-analysis provided the mean and standard deviation of pre- and post-intervention data for the main outcome.

### 3.2. Study Characteristics

The characteristics of the four studies included in the systematic review and meta-analysis can be found in Table 1.

Additionally, the details of the SSG-based and running-based HIIT programs can be found in Table 2. The included parallel studies involved 8 individual groups (4 SSG-based groups and 4 running-based HIIT groups) and 77 participants (n = 39 in SSG-based groups; n = 38 in running-based HIIT groups). Among the included studies, the smaller intervention lasted 4 weeks [30] and the longer 6 weeks [29,31]. Three of the interventions had two sessions per week [28,29,30], while one [31] had three sessions per week. The total number of sessions ranged between a minimum of 8 [30] and a maximum of 18 [31].

### 3.3. Methodological Quality

All the included studies were classified with 18 points (Table 3).

### 3.4. SSG vs. Running-Based HIIT Interventions on Repeated-Sprint Ability

A summary of the included studies and results of RSA reported before and after SSG-based and running-based HIIT interventions are provided in Table 4.

Four studies provided data for RSA, involving four SSG-based and four HIIT-based groups (pooled *n* = 77). There was no significant difference between SSG-based compared to HIIT-based training interventions on the effect over RSA (ES = 0.30; 95% CI = −0.14 to 0.73; *p* = 0.181; *I*^2^ = 0.0%; Egger’s test *p* = 0.332; Figure 2). The relative weight of each study in the analysis ranged from 24.2% to 26.5% (the size of the plotted squares reflects the statistical weight of each study).

The within-group analysis revealed no significant effect of SSG-based training interventions on RSA (ES = −0.23; 95% CI = −1.40 to 0.94; *p* = 0.697; *I*^2^ = 93.8%; Egger’s test *p* = 0.695; Figure 3). The relative weight of each study in the analysis ranged from 23.4% to 25.9%.

The within-group analysis revealed no significant effect of HIIT-based training interventions on RSA (ES = 0.08; 95% CI = −1.17 to 1.33; *p* = 0.899; *I*^2^ = 94.0%; Egger’s test *p* = 0.801; Figure 4). The relative weight of each study in the analysis ranged from 23.9% to 26.0%.

### 3.5. Adverse Effects

Among the included studies, none reported soreness, pain, fatigue, injury, damage, or adverse effects related to the SSG-based and running-based HIIT interventions.

## 4. Discussion

The purpose of this meta-analysis was to compare the effects of SSGs and HIIT-based interventions on soccer players’ RSA. In short, despite the small number of included studies, the results revealed no significant differences between the two types of training; neither type of training was found to significantly affect RSA.

SSGs are drill-based activities that fall within the scope of running-based HIIT training. The difference between SSGs and running-based HIIT is that SSGs are performed using the dynamics of the game (two teams and one ball). In a well-known pair of published articles [8,32], HIIT training was classified (based on training regimen) into several types: short-interval, long-interval, repeated sprint training, sprint interval training, and game-based training (which includes SSGs) [8].

Typically, SSGs are prescribed as a part of long-interval regimens (2–4 min of high-intensity, non-maximal-intensity exercise). Among the studies included in this meta-analysis, SSG duration varied between 45 s [30] and 4 min [29], with the number of sets ranging between 2 and 10 (2 sets in longer-duration cases and 10 sets in minimum-duration cases). Additionally, formats of play varied between two vs. two [28,30] and four vs. four [31] games, played on a minimum field size of 75 m^2^ [28] and a maximum field size of 100 m^2^ [30] per player (field lengths ranged between 18 and 50 m). However, smaller formats of play and pitch sizes imply restricted high-intensity running demands (e.g., high-speed running and sprinting) [33,34], as well as high variability in the stimuli [35,36]. Therefore, this would be expected to promote the favorable effects of running-based HIIT on RSA. However, no significant differences were found between SSG and running-based HIIT.

The absence of significant differences between groups might be related to the range of running-based HIIT. Among the training regimens, one involved short intervals (15 s–15 s work–rest) [28], one involved long intervals (4 min–4 min work–rest) [31], one involved repeated sprint training (40 m all-out, interspaced by 20 s rest) [29], and one involved sprint interval training (30 s all-out interspaced by 150 s rest) [30]. In 50% of the included articles, RSA was improved by HIIT more significantly than by SSG [28,30]. Of these two articles, one involved similar regimens of training for the HIIT and SSG groups [28], while the other applied different regimens (one regimen of speed production with a 1:5 work-to-rest ratio favoring the running-based HIIT and one of speed maintenance with a 1:1 ratio favoring SSG) [30]. Therefore, it is difficult to compare the effects of the two training types. In a recent systematic review with meta-analysis about HIIT in soccer [37], it was also observed that different HIIT training regimens did not vary in their ability to improve RSA, although there were greater expectations that sprint interval training or repeated sprint training would be more appropriate to benefit RSA.

Interestingly, the within-group changes also revealed no significant effects of training interventions in RSA. This was somewhat unexpected, mainly in the running-based HIIT group. The particularly detrimental results obtained in a long-interval HIIT intervention [31] could explain these results (Figure 4). However, considering that RSA depends on several other factors (e.g., energy supply, hydrogen accumulation, and muscle activation) [13], interventions made alone (e.g., short intervals and long intervals) may not be as effective as when they are combined with other methods (e.g., resistance training, sprinting training). Concurrent training might be worth exploring since RSA benefits from lower-limb power for change-of-direction and maximal speed as well as a good energy supply. For example, a study comparing concurrent training (eccentric overload and HIIT) with HIIT by itself in soccer players showed benefits in players who underwent concurrent training [38].

One of the limitations of the current systematic review and meta-analysis is that only English articles from the Web of Science, Scopus, SPORTDiscus, and PubMed were included, thus potentially overlooking other relevant publications. Another limitation is the reduced number of included articles. However, this serves to highlight the need for more research on this topic. For instance, the absence of research on women and professional players was apparent. The results might change based on such moderators or other factors, such as baseline levels or even the volume of training completed beyond the interventions. Future research on this topic should apply the same training regimen to SSGs and running-based HIIT to homogenize the methodological process. Future research should also account for the responding profile of players to determine which type of profile is more responsive to the interventions.

As for practical applications, this meta-analysis highlights the importance of including complementary training methods that may help to develop RSA. Among other training regimens, combining SSG with running-based HIIT [39,40] or with strength/power training [41] might promote neuromuscular stimuli support improvements in RSA. Such research is worthwhile since previous findings have consistently revealed the beneficial effects of SSGs and running-based HIIT in aerobic performance [10,42].

## 5. Conclusions

The current meta-analytical comparison revealed no significant changes in the effects of SSG-based and running-based HIIT interventions on soccer players’ RSA. Additionally, among the included parallel studies, the within-group analysis revealed no significant improvements after SSG or running-based HIIT interventions. Despite the limited number of studies included in the present analysis, the findings should be carefully considered as practical applications. Specifically, the results indicate that complementary training methods (e.g., strength/power training, combined interventions) could help to improve RSA due to their multifactor-dependent quality. Finally, more research comparing SSG and running-based HIIT is needed; no studies on women or professional players were found in the present analysis.

## Figures and Tables

**Figure 1 ijerph-18-02781-f001:**
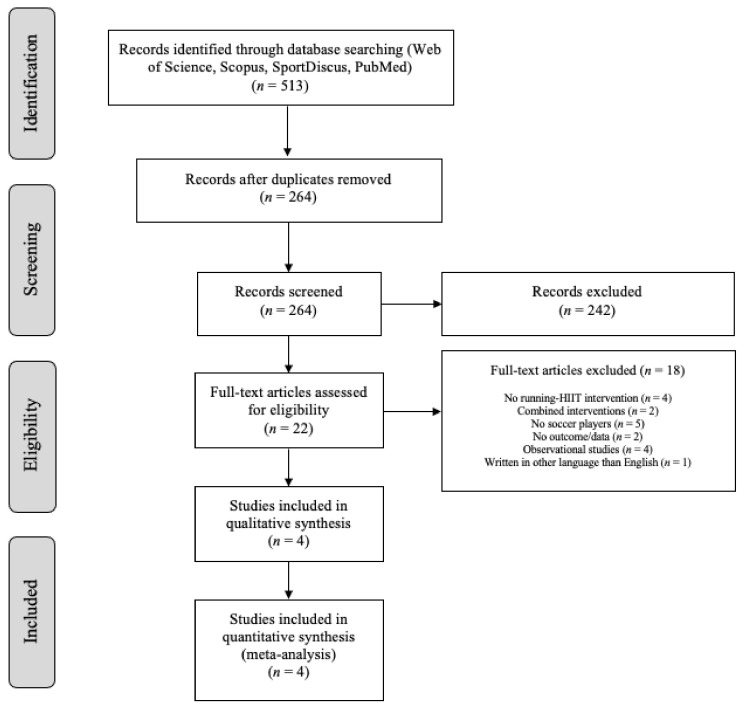
Preferred Reporting Items for Systematic Reviews and Meta-Analyses (PRISMA) flow diagram highlighting the selection process for the studies included in the systematic review and meta-analysis. HIIT: high-intensity interval training

**Figure 2 ijerph-18-02781-f002:**
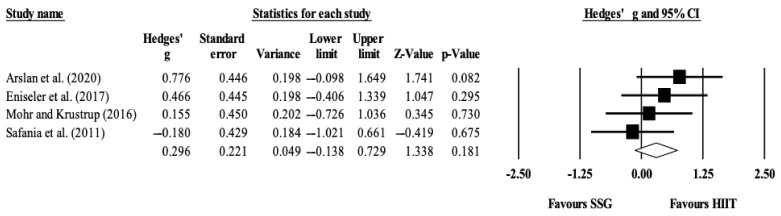
Forest plot of between-mode effect sizes (Hedges’ *g*) with 95% confidence intervals (CIs) in repeated sprint ability.

**Figure 3 ijerph-18-02781-f003:**
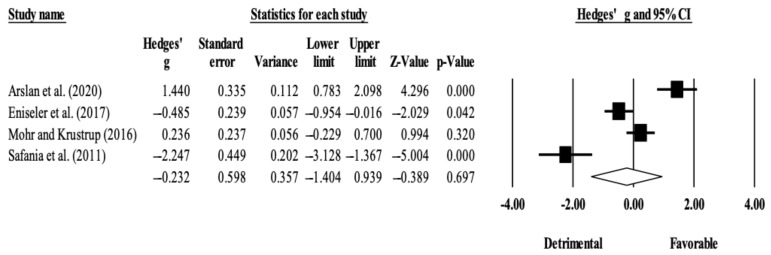
Forest plot of within-mode (SSG) effect sizes (Hedges’ *g*) with 95% confidence intervals (CIs) in repeated sprint ability.

**Figure 4 ijerph-18-02781-f004:**
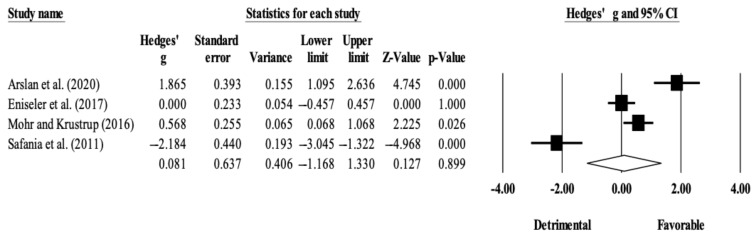
Forest plot of within-mode (HIIT) effect sizes (Hedges’ *g*) with 95% confidence intervals (CIs) in repeated sprint ability.

**Table 1 ijerph-18-02781-t001:** Characteristics of the included studies and outcomes extracted.

Study	Mean Age (Yo)	Sex	CL	Design	Tests Used in the Original Studies	Measure Extracted from the Tests in the Original Studies
Arslan et al. [28]	14.2	Men	Y	Parallel	6 × (2 × 15-m)/20 s recovery	RSA total (sum of all sprints)
Eniseler et al. [29]	16.9	Men	Y	Parallel	6 × (2 × 20-m)/20 s recovery	RSA mean time (mean of all sprints)
Mohr and Krustrup [30]	19	Men	U	Parallel	5 × 30-m/25 s recovery	RSA mean time (mean of all sprints)
Safania et al. [31]	15.7	Men	Y	Parallel	6 × 35-m/25 s recovery	Average power (mean of all sprints)

Yo: years old; CL: competitive level; Y: youth; U: university-level; s: seconds; m: meters; RSA: repeated-sprint ability

**Table 2 ijerph-18-02781-t002:** Characteristics of small-sided game (SSG)-based programs in the included studies.

Study	Intervention	Duration (w)	d/w	Total Sessions	SSG Formats	SSG Pitch Dimension(Length × Width)	SSG Area per Player (m^2^)	Sets	Reps	Recovery between Sets (Duration)	Recovery between Sets (Intensity)	Work Duration	Work Intensity	Between Reps Duration	Relief Intensity
Arslan et al. [28]	SSG	5	2	10	2 vs. 2	20 × 15-m	75 m^2^	2	2	2 min	-	2.5–4.5 min	NR	-	Passive
Eniseler et al. [29]	SSG	6	2	12	3 vs. 3	18 × 30-m	90 m^2^	4	-	4 min	-	3 min	90–95% HRmax	-	Passive
Mohr and Krustrup [30]	SSG	4	2	8	2 vs. 2	20 × 20-m	100 m^2^	8–10	-	45 s	-	45 s	NR	-	NR
Safania et al. [31]	SSG	6	3	18	2 vs. 2 to 4 vs. 4	10 × 15 to 40 × 50-m	NR	4	-	3 min	-	4 min	NR	-	NR
Arslan et al. [28]	HIIT	5	2	10	-	-	-	2	12–20	NR	NR	15 s	90–95% V_IFT_	15 s	Passive
Eniseler et al. [29]	HIIT	6	2	12	-	-	-	3	6	4 min	NR	40-m	All-out	20 s	Passive
Mohr & Krustrup [30]	HIIT	4	2	8	-	-	-	-	8–10	-	-	30 s	All-out	150 s	NR
Safania et al. [31]	HIIT	6	3	18	-	-	-	-	4	-	-	4 min	70–95% HRmax	3 min	NR

w: weeks; d/w: days per week; NR: not reported; m: meters; s: seconds; min: minutes; V_IFT_: maximal velocity at 30–15 Intermittent Fitness Test; IAT: individual anaerobic threshold; HRmax: maximal heart rate; Passive: passive recovery.

**Table 3 ijerph-18-02781-t003:** Methodological index for non-randomized studies (MINORS).

Study	N.º1 *	N.º2	N.º3	N.º4	N.º5	N.º6	N.º7	N.º8	N.º9	N.º10	N.º11	N.12	Total **
Arslan et al. [28]	2	1	2	2	0	2	2	0	1	2	2	2	18
Eniseler et al. [29]	2	1	2	2	0	2	2	0	1	2	2	2	18
Mohr and Krustrup [30]	2	1	2	2	0	2	2	0	1	2	2	2	18
Safania et al. [31]	2	1	2	2	0	2	2	0	1	2	2	2	18

*: MINORS scale items number; N.º1: A clear study aim; N.º2: Inclusion of consecutive patients; N.º3: Prospective collection of data; N.º4: Endpoints appropriate to the aim of the study; N.º5: Unbiased assessment of the study endpoint; N.º6: Follow-up period appropriate to the aim of the study; N.º7: Loss to follow-up less than 5%; N.º8: Prospective calculation of the study size; N.º9: An adequate control group; N.º10: Contemporary groups; N.º11: Baseline equivalence of groups; N.º12: Adequate statistical analyses; **: the total number of points from a possible maximal of 24.

**Table 4 ijerph-18-02781-t004:** Summary of the included studies and results of repeated sprint ability before and after SSG-based and running-based high-intensity interval training (HIIT) intervention.

Study	Intervention	*n*	Before Mean ± SD	After Mean ± SD	Before−After (Δ%)
Arslan et al. [28]	SSG	10	37.8 ± 1.5	35.6 ± 1.2	−5.8
Eniseler et al. [29]	SSG	10	7.12 ± 0.17	7.22 ± 0.20	1.4
Mohr and Krustrup [30]	SSG	9	4.41 ± 0.07	4.35 ± 0.22	−1.4
Safania et al. [31]	SSG	10	309.0 ± 39.0	220.0 ± 24.0	−28.8
Arslan et al. [28]	HIIT	10	38.2 ± 1.7	34.9 ± 1.5	−8.6
Eniseler et al. [29]	HIIT	9	7.13 ± 0.17	7.13 ± 0.21	0.0
Mohr and Krustrup [30]	HIIT	9	4.45 ± 0.05	4.36 ± 0.14	−2.0
Safania et al. [31]	HIIT	10	291.0 ± 38.0	207.0 ± 29.0	−28.9

*n*: number of participants per group; SD: standard deviation; SSG: small-sided game; HIIT: high-intensity interval training

## Data Availability

Not applicable.

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
