# Peer review of "A Meta-Analytical Comparison of the Effects of Small-Sided Games vs. Running-Based High-Intensity Interval Training on Soccer Players’ Repeated-Sprint Ability"

_ijerph, 2021, doi:10.3390/ijerph18052781_

Round 1

Reviewer 1 Report

The abstract is well outlined and presents the study objectives, the main results, as well as the main conclusions, although there are no significant differences, the authors suggest complementary training may be performed to improve the capacity of SSGs and HIIT. It also suggests that different forms of HIIT can be used because of the range of opportunities that such training encompasses.

The introduction is organized from the general to the specific. The authors were careful to try to find information on the current state of the art of the subject studied. They present a range of studies where they intend to analyze the possibility of finding justification in the intervention of the SSG. At the same time understand whether drill-based exercises can develop RSA to a similar extent as other forms of exercise (running-based HIIT) and if they are significantly beneficial for soccer players. Such an understanding (SSG vs. running-based HIIT) will help to define a practical application for the soccer field.

In my view, science and research only make sense if they are applied in practice.  We agree with this permis.

When we talk about Materials and Methods, the authors followed all the essential steps in the elaboration of the study objective. Permits, steps, prescribers, Data Extraction, Data items, Assessment of Methodological Quality, Summary Measures, Synthesis of Results, Risk of Bias Across Studies and finally Study Identification and Selection. The figure 1 is a demonstration of the drawing done with all the steps performed.

In presenting the results, the authors were able to have rigor, criteria and cohesion in the presentation of the 4 tables according to the themes studied.

Although there are no significant differences in the comparison of the two themes studied. Figures 2 and 3 are illustrative relative weight of each study in the analysis ranged from 23.4% to 25.9%. and The relative weight of each study in the analysis ranged from 23.9% to 26.0%.

Finally, as we can be seen, among the included studies, none reported soreness, pain, fatigue, injury, damage or adverse effects related to the SSG-based and running-based HIIT interventions.

In the passage to one of the key moments of the study in question, namely the discussion and its practical application, it seems to us to take into account the importance of these studies.

In short, despite the small number of included studies, the results revealed no significant differences between the two types of training; neither type of training was found to significantly affect RSA.

SSGs are drill-based activities that fall within the range of running-based HIIT training. The difference between SSGs and running-based HIIT is that SSGs are performed using the dynamics of the game (two teams and one ball).

In a well-known pair of published articles Buchheit et al 2013 and Buchheit et al 2013a, HIIT training was classified (based on training regimen) into several types: short-interval, long-interval, repeated sprint training, sprint interval training, and game-based training (which include SSGs).

According the authors, it would be expectable to assist to a favorable effect of running-based HIIT on the RSA. However, no significant differences were found between SSG and running-based HIIT. When However, smaller formats of play and pitch sizes imply restricted high-intensity running demands.

Also similar regimens of training for the HIIT and SSG groups, while the other applied different regimens (one of speed production with a 1:5 work-to-rest ratio favoring the running-HIIT and one of speed maintenance with a 1:1 ratio favoring SSG).

Therefore, it is difficult to compare the effects of the two training types. The within-group changes also revealed no significant effects of training interventions in RSA.

Another study comparing concurrent training (eccentric overload and HIIT) with just HIIT in soccer players showed benefits in players who underwent concurrent training. Concurrent training might be worth exploring since RSA benefits from lower-limb power for change-of-direction and maximal speed as well as a good energy supply.

The authors were careful to include the possible limitations of the study.

Leaving a possibility of future exploration in order to reduce the error or possible limitations:

  • One of the limitations of the current systematic review and meta-analysis is that only English articles from the Web of Science, Scopus, SPORTDiscus, and PubMed were included, thus potentially overlooking other relevant publications.

  • Another limitation is the reduced number of included articles. However, such a fact highlights the need for more research on this topic. For instance, the absence of research in women and professionals was apparent.

Another concern in this study by the authors, was the future research:

  • on this topic should apply the same training regimen to SSGs and running-based HIIT to homogenize the methodological process.

  • research should also control for the responding profile of players to determine which type of profile is more responsive to the interventions.

Regarding the practical application of the studies, despite the lack of differences, this study is something new that must be taken into account:

  • highlights the importance of including complementary training methods that may help to develop RSA.

  • combining SSG with running-based HIIT or with strength/power training might guarantee an improvement in the neuromuscular stimuli support improvements in RSA.

  • Such research is worthwhile since previous findings have consistently revealed the beneficial effects of SSGs and running-based HIIT in aerobic performance.

Finally, since the conclusions summarize the applied research, we arrived at the study's revelations or not and what they represent, for researchers, coaches and the phenomenon of the studied SSG theme:

  • The current meta-analytical comparison revealed no significant changes in the effects of SSG-based and running-based HIIT interventions on soccer players’ RSA.

  • within-group analysis revealed no significant improvements after SSG or running-based HIIT interventions.

  • Despite the limited number of studies included in the present analysis, the findings should be carefully considered as practical applications.

  • The results indicate that complementary training methods (e.g., strength/power training; combined interventions) could help to improve RSA due to their multifactorial-dependent quality.

Despite the demand of those who research and study, it is not possible to reach a definite and unique conclusion, so the need to continue looking so that we can work towards more discoveries and new indicators by all the stakeholders:

  • More research is needed comparing SSG and running-based HIIT;
  • No studies on women or professionals were found in the present analysis.

The bibliography is in accordance with the theme studied and updated regarding the last years, more precisely, the last 5 - 10 years. They are in accordance with the journal's standards.

The theme is interesting, although most of the samples are mostly male and young and university students.

These are the same focus of this type of intervention, both in training and in teaching. Although at the senior level and in many sports they are a way of approaching learning and micro intervention strategy, to be able to resolve in a more macro format in competition. Often used in Team sports.

Author Response

REVIEWER 1

The abstract is well outlined and presents the study objectives, the main results, as well as the main conclusions, although there are no significant differences, the authors suggest complementary training may be performed to improve the capacity of SSGs and HIIT. It also suggests that different forms of HIIT can be used because of the range of opportunities that such training encompasses.

AUTHORS: DEAR REVIEWER, THANK YOU SO MUCH FOR YOUR COMMENTS AND SUGGESTIONS. THE CHANGES RELATED TO YOUR COMMENTS WERE HIGHLIGHTED IN GREEN.

The introduction is organized from the general to the specific. The authors were careful to try to find information on the current state of the art of the subject studied. They present a range of studies where they intend to analyze the possibility of finding justification in the intervention of the SSG. At the same time understand whether drill-based exercises can develop RSA to a similar extent as other forms of exercise (running-based HIIT) and if they are significantly beneficial for soccer players. Such an understanding (SSG vs. running-based HIIT) will help to define a practical application for the soccer field. In my view, science and research only make sense if they are applied in practice.  We agree with this permis.

AUTHORS: DEAR REVIEWER, THANK YOU SO MUCH FOR YOUR COMMENTS

When we talk about Materials and Methods, the authors followed all the essential steps in the elaboration of the study objective. Permits, steps, prescribers, Data Extraction, Data items, Assessment of Methodological Quality, Summary Measures, Synthesis of Results, Risk of Bias Across Studies and finally Study Identification and Selection. The figure 1 is a demonstration of the drawing done with all the steps performed.

AUTHORS: DEAR REVIEWER, THANK YOU SO MUCH FOR YOUR COMMENTS

In presenting the results, the authors were able to have rigor, criteria and cohesion in the presentation of the 4 tables according to the themes studied.

AUTHORS: DEAR REVIEWER, THANK YOU SO MUCH FOR YOUR COMMENTS

Although there are no significant differences in the comparison of the two themes studied. Figures 2 and 3 are illustrative relative weight of each study in the analysis ranged from 23.4% to 25.9%. and The relative weight of each study in the analysis ranged from 23.9% to 26.0%.

AUTHORS: DEAR REVIEWER, THANK YOU SO MUCH FOR YOUR COMMENTS

Finally, as we can be seen, among the included studies, none reported soreness, pain, fatigue, injury, damage or adverse effects related to the SSG-based and running-based HIIT interventions.

AUTHORS: DEAR REVIEWER, THANK YOU SO MUCH FOR YOUR COMMENTS

In the passage to one of the key moments of the study in question, namely the discussion and its practical application, it seems to us to take into account the importance of these studies.

AUTHORS: DEAR REVIEWER, THANK YOU SO MUCH FOR YOUR COMMENTS

In short, despite the small number of included studies, the results revealed no significant differences between the two types of training; neither type of training was found to significantly affect RSA.

AUTHORS: DEAR REVIEWER, THANK YOU SO MUCH FOR YOUR COMMENTS

SSGs are drill-based activities that fall within the range of running-based HIIT training. The difference between SSGs and running-based HIIT is that SSGs are performed using the dynamics of the game (two teams and one ball).

AUTHORS: DEAR REVIEWER, THANK YOU SO MUCH FOR YOUR COMMENTS

In a well-known pair of published articles Buchheit et al 2013 and Buchheit et al 2013a, HIIT training was classified (based on training regimen) into several types: short-interval, long-interval, repeated sprint training, sprint interval training, and game-based training (which include SSGs). According the authors, it would be expectable to assist to a favorable effect of running-based HIIT on the RSA. However, no significant differences were found between SSG and running-based HIIT. When However, smaller formats of play and pitch sizes imply restricted high-intensity running demands.

AUTHORS: DEAR REVIEWER, THANK YOU SO MUCH FOR YOUR COMMENTS

Also similar regimens of training for the HIIT and SSG groups, while the other applied different regimens (one of speed production with a 1:5 work-to-rest ratio favoring the running-HIIT and one of speed maintenance with a 1:1 ratio favoring SSG).

AUTHORS: DEAR REVIEWER, THANK YOU SO MUCH FOR YOUR COMMENTS

Therefore, it is difficult to compare the effects of the two training types. The within-group changes also revealed no significant effects of training interventions in RSA.

AUTHORS: DEAR REVIEWER, THANK YOU SO MUCH FOR YOUR COMMENTS

Another study comparing concurrent training (eccentric overload and HIIT) with just HIIT in soccer players showed benefits in players who underwent concurrent training. Concurrent training might be worth exploring since RSA benefits from lower-limb power for change-of-direction and maximal speed as well as a good energy supply.

AUTHORS: DEAR REVIEWER, THANK YOU SO MUCH FOR YOUR COMMENTS

The authors were careful to include the possible limitations of the study.

AUTHORS: DEAR REVIEWER, THANK YOU SO MUCH FOR YOUR COMMENTS

Leaving a possibility of future exploration in order to reduce the error or possible limitations:

  • One of the limitations of the current systematic review and meta-analysis is that only English articles from the Web of Science, Scopus, SPORTDiscus, and PubMed were included, thus potentially overlooking other relevant publications.
  • Another limitation is the reduced number of included articles. However, such a fact highlights the need for more research on this topic. For instance, the absence of research in women and professionals was apparent.

 AUTHORS: DEAR REVIEWER, THANK YOU SO MUCH FOR YOUR COMMENTS

Another concern in this study by the authors, was the future research:

  • on this topic should apply the same training regimen to SSGs and running-based HIIT to homogenize the methodological process.

  • research should also control for the responding profile of players to determine which type of profile is more responsive to the interventions.

 AUTHORS: DEAR REVIEWER, THANK YOU SO MUCH FOR YOUR COMMENTS

Regarding the practical application of the studies, despite the lack of differences, this study is something new that must be taken into account:

  • highlights the importance of including complementary training methods that may help to develop RSA.
  • combining SSG with running-based HIIT or with strength/power training might guarantee an improvement in the neuromuscular stimuli support improvements in RSA.
  • Such research is worthwhile since previous findings have consistently revealed the beneficial effects of SSGs and running-based HIIT in aerobic performance.

 AUTHORS: DEAR REVIEWER, THANK YOU SO MUCH FOR YOUR COMMENTS

Finally, since the conclusions summarize the applied research, we arrived at the study's revelations or not and what they represent, for researchers, coaches and the phenomenon of the studied SSG theme:

  • The current meta-analytical comparison revealed no significant changes in the effects of SSG-based and running-based HIIT interventions on soccer players’ RSA.
  • within-group analysis revealed no significant improvements after SSG or running-based HIIT interventions.
  • Despite the limited number of studies included in the present analysis, the findings should be carefully considered as practical applications.
  • The results indicate that complementary training methods (e.g., strength/power training; combined interventions) could help to improve RSA due to their multifactorial-dependent quality.

 AUTHORS: DEAR REVIEWER, THANK YOU SO MUCH FOR YOUR COMMENTS

Despite the demand of those who research and study, it is not possible to reach a definite and unique conclusion, so the need to continue looking so that we can work towards more discoveries and new indicators by all the stakeholders:

  • More research is needed comparing SSG and running-based HIIT;
  • No studies on women or professionals were found in the present analysis.

 AUTHORS: DEAR REVIEWER, THANK YOU SO MUCH FOR YOUR COMMENTS

The bibliography is in accordance with the theme studied and updated regarding the last years, more precisely, the last 5 - 10 years. They are in accordance with the journal's standards.

AUTHORS: DEAR REVIEWER, THANK YOU SO MUCH FOR YOUR COMMENTS

The theme is interesting, although most of the samples are mostly male and young and university students.

AUTHORS: DEAR REVIEWER, THANK YOU SO MUCH FOR YOUR COMMENTS

These are the same focus of this type of intervention, both in training and in teaching. Although at the senior level and in many sports they are a way of approaching learning and micro intervention strategy, to be able to resolve in a more macro format in competition. Often used in Team sports.

AUTHORS: DEAR REVIEWER, THANK YOU SO MUCH FOR YOUR COMMENTS

Reviewer 2 Report

The authors review the effect of small-sided games and high intensity interval training (based on running). I supposed already before reading the main part of the paper that the training based on small-sides games and HIIT are in some sense similar in intensity so it should not be a huge difference in effect. Anyway, I find the topic interesting and worth further investigating. 

In my opinion I do not see any drawbacks of the manuscript. All the sections (methods, results, conclusions) are well-prepared. I would encourage authors to examine sprint interval training (SIT). It is supposed that SIT is more effective and time-efficient than HIIT. I think SIT would have a significant improvement on RSA. 

The one element might be improved. Please explain abbreviation already in the  abstract. HITT is commonly known, but RSA and SSG is not. That may help a reader to decide about reading just from looking at the abstract. 

Author Response

REVIEWER 2

The authors review the effect of small-sided games and high intensity interval training (based on running). I supposed already before reading the main part of the paper that the training based on small-sides games and HIIT are in some sense similar in intensity so it should not be a huge difference in effect. Anyway, I find the topic interesting and worth further investigating. 

AUTHORS: DEAR REVIEWER, THANK YOU SO MUCH FOR YOUR COMMENTS AND SUGGESTIONS. WE HAVE CHANGED THE MANUSCRIPT BASED ON YOUR SUGGESTIONS. THE CHANGES RELATED TO YOUR COMMENTS WERE HIGHLIGHTED IN YELLOW.

In my opinion I do not see any drawbacks of the manuscript. All the sections (methods, results, conclusions) are well-prepared. I would encourage authors to examine sprint interval training (SIT). It is supposed that SIT is more effective and time-efficient than HIIT. I think SIT would have a significant improvement on RSA.

AUTHORS: DEAR REVIEWER, THANK YOU. WE HAVE ADDED AN EXTENSION IN THE DISCUSSION REGARDING THIS TOPIC.

The one element might be improved. Please explain abbreviation already in the abstract. HITT is commonly known, but RSA and SSG is not. That may help a reader to decide about reading just from looking at the abstract. 

AUTHORS: DEAR REVIEWER, THANK YOU. WE HAVE ADDED THE FULL NAMES IN THE ABSTRACT.